# Highly Conserved Elements and Chromosome Structure Evolution in Mitochondrial Genomes in Ciliates

**DOI:** 10.3390/life7010009

**Published:** 2017-02-27

**Authors:** Roman A. Gershgorin, Konstantin Yu. Gorbunov, Oleg A. Zverkov, Lev I. Rubanov, Alexandr V. Seliverstov, Vassily A. Lyubetsky

**Affiliations:** 1Institute for Information Transmission Problems of the Russian Academy of Sciences (Kharkevich Institute), Bolshoy Karetny per. 19, build.1, Moscow 127051, Russia; gershr@mail.ru (R.A.G.); gorbunov@iitp.ru (K.Y.G.); rubanov@iitp.ru (L.I.R.); slvstv@iitp.ru (A.V.S.); lyubetsk@iitp.ru (V.A.L.); 2Faculty of Mechanics and Mathematics, Lomonosov Moscow State University, Leninskiye Gory 1, Main Building, Moscow 119991, Russia

**Keywords:** Ciliophora, mitochondria, highly conserved elements, proteins clustering, chromosome structure, evolution

## Abstract

Recent phylogenetic analyses are incorporating ultraconserved elements (UCEs) and highly conserved elements (HCEs). Models of evolution of the genome structure and HCEs initially faced considerable algorithmic challenges, which gave rise to (often unnatural) constraints on these models, even for conceptually simple tasks such as the calculation of distance between two structures or the identification of UCEs. In our recent works, these constraints have been addressed with fast and efficient solutions with no constraints on the underlying models. These approaches have led us to an unexpected result: for some organelles and taxa, the genome structure and HCE set, despite themselves containing relatively little information, still adequately resolve the evolution of species. We also used the HCE identification to search for promoters and regulatory elements that characterize the functional evolution of the genome.

## 1. Introduction

ATP and other compounds are synthesized in mitochondria [1]. Generally, many eukaryotes living under anaerobic conditions either lack mitochondria [2], or contain mitochondrial remnants such as hydrogenosomes or mitosomes. For example, *Nyctotherus ovalis*, anaerobic, lives in the hindgut of the cockroaches *Periplaneta americana* and *Blaberus* sp. [3]; its mitochondria generate hydrogen [4]. The role of mitochondria varies between different organisms, and is reflected in the size of the mitochondrial genomes. Parasitic apicomplexans have extremely small mitochondrial genomes coding for only three proteins and short rRNA fragments [5].

The ciliates (Ciliophora) include parasitic *Ichthyophthirius multifiliis* which causes death in many freshwater fish species reared in aquaria, fish farms, and aquacultures [6]. Mitochondria of ciliates can serve as targets for therapeutic intervention in parasitic diseases, and analysis of the structure and evolution of their genomes as well as the regulation of their gene expression can be of practical importance, in particular in veterinary medicine, e.g., for organization and veterinary care in fish hatcheries.

We analyzed the mitochondrial genomes in Ciliophora. The phylum Apicomplexa as well as the phylum Ciliophora belong to the superphylum Alveolata. We considered genera that belong to three classes: Armophorea (*Nyctotherus*), Oligohymenophorea (*Ichthyophthirius*, *Paramecium*, and *Tetrahymena*), and Spirotrichea (*Moneuplotes*, *Oxytricha*). Twelve complete mitochondrial genomes considered here are listed at the beginning of Materials and Methods. Oligohymenophorea and Spirotrichea substantially differ [7]. Armophorea includes anaerobes but is closer to Spirotrichea than to Oligohymenophorea [4]. Many ciliates are free-living organisms. For example, *Moneuplotes minuta* cells can be collected in the Mediterranean Sea near Corsica [7]. Both *Moneuplotes minuta* and *Oxytricha trifallax* can be cultured in inorganic salt medium with *Chlamydomonas reinhardtii* or *Klebsiella* spp. as food sources. On the contrary, the ciliate *Ichthyophthirius multifiliis* is a pathogen of freshwater fish occurring in both temperate and tropical regions throughout the world [8]. It is less tolerant of salt than fish. Both *Tetrahymena* and *Paramecia* are free-living ciliate protozoa. *Tetrahymena* are common in freshwater ponds. *Paramecia* are widespread in freshwater, brackish, and marine environments and are often very abundant in stagnant basins and ponds. The endosymbionts of *Paramecium aurelia* are Gram-negative bacteria. Most of the endosymbionts produce toxins which kill sensitive strains of *Paramecia* [9].

The mitochondria considered here code for tens of proteins [4,7,8,9,10,11,12,13,14,15]. The functions of some of them remain unclear, and they relatively rapidly accumulate substitutions [16]. The mitochondrial chromosome is circular in *Ichthyophthirius* and linear in other species considered here [15,17]. In the mitochondria of *Tetrahymena*, *Moneuplotes*, and *Oxytricha*, most genes are transcribed in opposite directions from the middle of the linear chromosome. In contrast, most genes are transcribed in the same direction in the mitochondria of *Paramecium* and *Nyctotherus*.

The considered mitochondrial genomes are very compact. Genes form long operons with short non-coding regions; the coding regions can overlap in some cases. The order of genes differs between the classes considered, which makes the analysis of evolution of the chromosome structure alone a nontrivial task. The class Oligohymenophorea features relatively long non-coding regions upstream of the gene encoding apocytochrome b.

The goal of this report was confined to the application of the algorithm for the identification of highly conserved elements (HCEs) as well as the algorithm of chromosome structure evolution presented in [18,19] to the mitochondrial data of taxonomically distant species. In addition, the study of the statement formulated in the next paragraph was initiated as well as the analysis of the identified HCEs.

Traditionally, studies of species evolution to a large extent relied on the comparative analysis of genomic regions coding for rRNAs and proteins apart from the analysis of morphological characters. Later, analyses made use of regulatory elements and the structure of the genome as a whole. More recently, phylogenetic analyses start to incorporate ultraconserved elements (UCEs) and highly conserved elements (HCEs). Models of evolution of the genome structure and HCE initially faced considerable algorithmic challenges, which gave rise to (often unnatural) constraints on these models, even for conceptually simple tasks such as the calculation of distance between two structures or the identification of UCEs. These constraints are now being addressed with fast and efficient solutions with no constraints on the underlying models [18,19]. These approaches have led us to an unexpected result: at least for some organelles and taxa, the genome structure and HCE set, despite themselves containing relatively little information, still adequately resolve the evolution of species.

We also used the HCE identification to search for promoters and regulatory elements that characterize the functional evolution of the genome.

## 2. Results and Discussion

### 2.1. Highly Conserved Elements in Mitochondrial Genome in Ciliates (Ciliophora)

Our computer program based on the original algorithm [18] was used to identify highly conserved DNA elements referred to as HCEs. As a result, 393 HCEs have been identified and assigned unique numbers (see Appendix A). Figure 1 demonstrates the tree generated by RAxML [20] from a matrix with 12 rows and 393 columns showing the presence or absence in each mitochondrial genome of each HCE. Notice that this popular program deals with a matrix of ones and zeros, which distinguishes it from, e.g., PhyloBayes and the neighbor-joining method. RAxML implements the maximum likelihood method (ML). This tree is in good but naturally imprecise agreement with the species tree based on GenBank taxonomy. In particular, *Moneuplotes crassus* more commonly shared HCEs with *Oxytricha trifallax* than with *Moneuplotes minuta*, while *Paramecium aurelia* notably differed from *P. caudatum* by the HCE pattern.

Five HCEs have been found in Oligohymenophorea (assigned numbers 138, 234, 287, 290, and 315), neither overlapping with gene coding regions nor corresponding to RNA species described in Rfam. Four out of five of these HCEs have been found only within the *Tetrahymena* genus. The identified HCEs are described in Appendix A. Table 1 exemplifies six HCEs found in Oligohymenophorea.

**HCE 287** has been found only in four *Tetrahymena* species. It is located upstream of the rRNA large subunit (on the complementary strand). It can be involved in the regulation of transcription or in post-transcriptional modifications of rRNA.

**HCE 299**. The mitochondrial *nad2* and *nad7* genes have opposite orientations and close positions in Oligohymenophorea; each of them starts a long operon. The alignment of Nad2 amino acid sequences annotated in GenBank demonstrates that there are nearly no conserved positions at the N terminus. Conversely, the *nad7* genes are highly conserved, and their 5′ ends overlap with HCE 151 in *Ichthyophthirius multifiliis*, *Tetrahymena malaccensis*, *T. paravorax*, *T. pigmentosa*, *T. pyriformis*, and *T. thermophile*.

This suggests that the *nad2* gene overlaps the promoter upstream of *nad7*. HCE 299, containing a potential promoter, has been found within the *nad2* coding regions in *Tetrahymena malaccensis*, *T. pigmentosa*, *T. pyriformis*, and *T. thermophile*. The CATA sequence (boldfaced in Table 1) corresponds to the YRTA consensus of promoters in plant mitochondria [21].

**HCE 234** has been found in all *Tetrahymena* species between the *ymf76* and *ymf66* genes (both on the complementary strand). In four species, *T. malaccensis, T. paravorax, T. pigmentosa*, and *T. pyriformis*, HCE 234 is neighbored by HCE 290. The TGTA sequence (boldfaced in Table 1) corresponds to the YRTA consensus of promoters in plant mitochondria [21]. Analysis of potential promoters within HCE 290 and HCE 299 exposes a conserved motif, YRTAnnAATTY. However, the genes around HCE 290 are on the complementary strand.

**HCEs 138 and 315**. The *Tetrahymena* spp. *cob* gene coding for apocytochrome b is in an opposite orientation to the *ymf77* gene. The *Tetrahymena pyriformis* genome annotation indicates a PAL2 element between these genes close to *ymf77*, which is similar to the parasitic PAL2-1 element from the mitochondria of *Neurospora* and *Podospora*, a senescence factor in these fungi [22]. A conserved motif has been found in this intergenic region closer to the *cob* gene. It corresponds to HCE 138 found in a wide range of species including *Ichthyophthirius multifiliis* (two regions, both between pairs of the gene encoding the large subunit ribosomal RNA), *Tetrahymena malaccensis, T. paravorax, T. pigmentosa, T. pyriformis*, and *T. thermophila*. Different localization of HCE 138 in *Ichthyophthirius multifiliis* and *Tetrahymena* spp. confirms that this element is associated with the mobile element rather than with the gene.

The same genome region harbors HCE 315, which was found only in four *Tetrahymena* species. Three out of four of these elements contain the CGTA sequence corresponding to the YRTA consensus of promoters in plant mitochondria [21]. This can be a promoter of the operon starting with the *cob* gene. However, a nucleotide was substituted in this site in *T. pyriformis*.

HCE 315 has not been found in other Oligohymenophorea, which suggests the presence of another promoter upstream of the *cob* gene in them. Indeed, a potential promoter with a different sequence has been identified in *Ichthyophthirius multifiliis* and *Paramecium* spp.

Figure 2 shows the alignment of the 5′-leader sequences upstream of the *cob* gene in *Ichthyophthirius multifiliis* and *Paramecium* spp. The conserved region with low similarity to plant mitochondrial promoters is marked in grey; however, this region contains no YRTA site typical for such promoters [21]. The *cob* gene in these species is surrounded with other genes in the same DNA strand; however, the 5′-intergenic region of *cob* is relatively long.

### 2.2. Clustering of Proteins Encoded in Mitochondria in Ciliates

We used our algorithm [23] to divide the proteins encoded in mitochondria into clusters, presumable protein families. The obtained protein families are available at [24] as a database, which can be searched by protein phylogenetic profile. It should be noted that different clustering methods are also discussed in [25].

Thus, 550 proteins from 12 mitochondria were divided into 63 non-single-element (nontrivial) clusters and 109 single-element clusters (singletons). Most singletons are represented by proteins from *Oxytricha trifallax* and *Nyctotherus ovalis*.

Only one cluster including NADH dehydrogenase subunit 9 (Nad9) proteins contains paralogs. Specifically, two *Tetrahymena* species, *T. malaccensis* and *T. thermophila*, include very similar pairs of proteins YP_740744.1 (Nad9_1) and YP_740745.1 as well as (Nad9_2) NP_149392.1 (Nad9_1) and NP_149393.1 (Nad9_2), emerging from a recent duplication, presumably in their nearest common ancestor. Indeed, these species form a clade in two evolutionary trees discussed below, while they essentially form a polytomous group in the HCE-based tree (Figure 1). However, this conclusion can be refined. The proteins in each of the two pairs differ by a single position (specific for each pair), while the four proteins composing these pairs differ by 18 positions. Hence, it is more reasonable to propose independent duplications in these two species. The evolution of these paralogs was reconstructed by generating the tree of the Nad9 cluster using the PhyloBayes program (Figure 3), in particular demonstrating that each paralog is nearly equidistant from other proteins of the family. PhyloBayes implements commonly used Bayesian inference.

The size distribution of the clusters is shown in Figure 4; the number of proteins in each species in clusters and singletons is given in Table 2.

Finally, all clusters (39 in total) representing at least six species were selected. An alignment was generated for each of them using MUSCLE as described below in Materials and Methods. The trimAl program was then used to remove low-informative alignment columns. The alignments were concatenated into a single one with a total length of 8701 amino acids and the missing data ratio of 26%. RAxML was used to generate an evolutionary tree for the mitochondria of the species considered from this concatenated alignment; the tree was in a good agreement with the generally accepted taxonomy. Exactly the same tree has been generated by the PhyloBayes program (Figure 5). The tree has maximum support at all nodes (100% bootstrap values for RAxML and posterior probability of 1 for PhyloBayes). This is a common practice in tree building from protein data.

### 2.3. Evolution of Mitochondrial Chromosome Structure in Ciliates

The evolutionary tree of mitochondrial chromosome structures was generated from the distances between them, calculated using the chromosome structure model proposed in [19] and the program available at [26].

The resulting tree is shown in Figure 6. Each genus forms a clade in it. The Armophorea, Oligohymenophorea, and Spirotrichea classes also form clades. The close position of Armophorea and Spirotrichea on the tree is consistent with published data [4]. Thus, there is a largely good agreement between the HCE-based tree (Figure 1), the tree of proteins (Figure 5), the tree of chromosome structures (Figure 6), and the generally accepted taxonomy. Minor differences between the trees shown in Figure 1, Figure 4 and Figure 5 can be attributed to the small size of the mitochondrial genomes and the corresponding relative scarcity of data.

### 2.4. Reconstruction of Mitochondrial Chromosome Structure in Ciliates

The results of the reconstruction of the mitochondrial chromosome structures in ciliates at the internal nodes of the tree generated by the method described in [19] are presented in Appendix A. The left column of the table lists the tree leaf designated as (*l*) or terminal (according to Figure 6) leaves descending from the considered internal node. The middle column contains the order of genes on the chromosome at the corresponding node; *L* and *C* indicate linear and circular chromosomes, respectively; the asterisk preceding the gene name indicates that it is encoded in the complementary strand. The second chromosome (if any) in the species corresponding to the node begins a new line. The chromosomes at the leaves served as initial data for our algorithm. The right column lists evolutionary events that occurred at the edge incident to the considered node.

## 3. Materials and Methods

Complete mitochondrial genomes were extracted from GenBank for the following species: *Ichthyophthirius multifiliis* (NC_015981), *Paramecium aurelia* (NC_001324), *P. caudatum* (NC_014262), *Tetrahymena malaccensis* (NC_008337), *T. paravorax* (NC_008338), *T. pigmentosa* (NC_008339), *T. pyriformis* (NC_000862), and *T. thermophila* (NC_003029). The same source was used to extract four incomplete mitochondrial genomes of *Moneuplotes minuta* (GQ903130), *M. crassus* (GQ903131), *Nyctotherus ovalis* (GU057832), and *Oxytricha trifallax* (*Sterkiella histriomuscorum*; JN383843).

The search for HCEs was performed using the algorithm based on the dense subgraph identification described elsewhere [18]. A similar method relying on pseudo-boolean programming is discussed in [27]. The following parameters were used: key length, 8; minimum word length, 24; maximum cost of difference between words, 3.1; minimum overlap length of merged words, 20; number of consecutive deletions, 0; deletion cost, 2.1; maximum key repeat count, 1000; maximum word compaction ratio, 2.2; minimum number of different words in a word and a key, 4 and 3, respectively.

The HCE-based tree in Figure 1 was generated using the RAxML program [20] from a matrix with 12 rows and 393 columns with the cells containing 1 or 0 to indicate the presence or absence of a given HCE in the mitochondrial genome of a given species, respectively. Maximum likelihood search followed by rapid bootstrapping was performed in RAxML v. 8.2.4 with the binary substitution model and maximum likelihood estimate for the base frequencies; number of bootstrap replicates was limited to 300 using the frequency-based criterion.

The distances between chromosome structures as well as the reconstruction of chromosome rearrangements were obtained by the methods described elsewhere [19,28,29]. The default operation costs were used, specifically: the linear variant and double-cut-and-paste, 1.2; sesqui-cut-and-paste, 1.1; *a*-edge insertion and *b*-edge deletion, 1; *b*-edge insertion or *a*-edge deletion, 0.9; deletion of special *a*-edges, 2.0; deletion of special *b*-edges, 2.5. The unrooted tree shown in Figure 6 was generated from the distances between chromosome structures using the neighbor-joining method [30].

Proteins were clustered using the method described and tested in [23,31,32,33,34]. BLAST hit threshold *E* = 0.001 and the most relaxed values for additional parameters: *L* = 0, *H* = 1, and very high *p* were used for clustering.

Protein alignment was performed by the MUSCLE program v. 3.8.31 [35] with default settings. Then, the trimAl program v. 1.2 [36] was used to remove positions with more than 50% gaps or with the similarity below 0.001. RAxML [20] and PhyloBayes v. 4.1 [37,38,39] with the MtZoa mitochondrial model [40] were used for tree generation. In the case of PhyloBayes, four chains ran in parallel for more than a thousand cycles. Upon convergence of likelihood values, alpha parameter, and tree length of the four chains, the discrepancy of bipartition frequencies between all chains was equal to zero (as shown by bpcomp utility in the PhyloBayes package). The first hundred cycles of each chain were discarded as burn-in and the majority rule consensus tree containing the posterior probabilities was calculated from the remaining trees of all chains. Both algorithms yielded the same tree. Trees with the same topology were also generated using more general CAT + GTR + Γ model in PhyloBayes and GTR + Γ model in RAxML.

Potential binding sites of transcription factors and promoters were identified using the method described elsewhere [41,42]. This method was used previously to identify binding sites of transcription factors in algal plastids [23,33,34].

GenBank gene annotations overlapping with HCEs were additionally checked against the Rfam database v. 12.1 [43].

## 4. Conclusions

At least for some organelles and taxa, the genome structure and HCE set, despite themselves containing relatively little information, still adequately describe the evolution of species. Indeed, the trees of HCEs, proteins, chromosome structures, and species are in agreement for the considered material. HCEs were found in mitochondrial genomes of ciliates (Ciliophora). Families of proteins encoded in mitochondria as well as the evolution of the chromosome structure were described in ciliate species. The data obtained were used to propose a method of combined application of our original methods to describe HCEs, protein families, and chromosome structures and eventually their evolution.

## Figures and Tables

**Figure 1 life-07-00009-f001:**
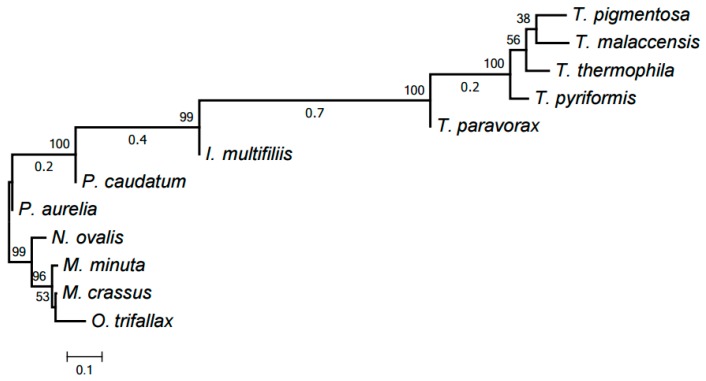
The tree of mitochondrial evolution generated using 393 HCEs identified by our algorithm. The tree was generated by the RAxML program based on a matrix with 12 rows and 393 columns, with the matrix cells containing 1 or 0 to indicate the presence or absence of a given HCE in the mitochondrial genome of a given species, respectively.

**Figure 2 life-07-00009-f002:**
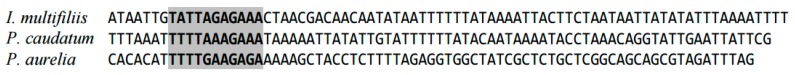
Alignment of 5′-leader sequences upstream of the cob gene.

**Figure 3 life-07-00009-f003:**
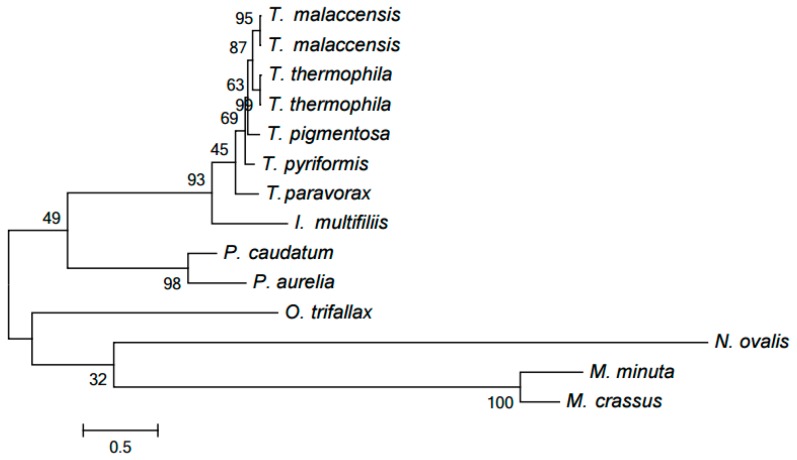
Tree of NADH dehydrogenase subunit 9 (Nad9) family according to our clustering. The tree was generated by PhyloBayes.

**Figure 4 life-07-00009-f004:**
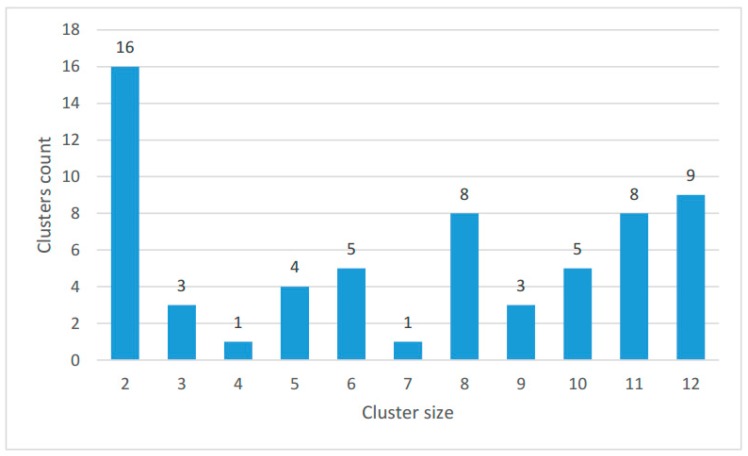
Size distribution of the clusters. The bar height shows the number of clusters including proteins from the number of species indicated on the abscissa.

**Figure 5 life-07-00009-f005:**
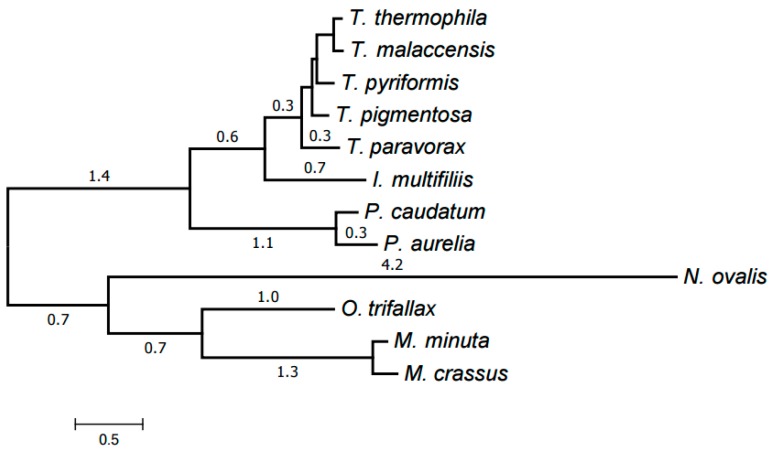
Evolutionary tree of mitochondria generated by PhyloBayes using the identified protein families. All nodes have the maximum support values.

**Figure 6 life-07-00009-f006:**
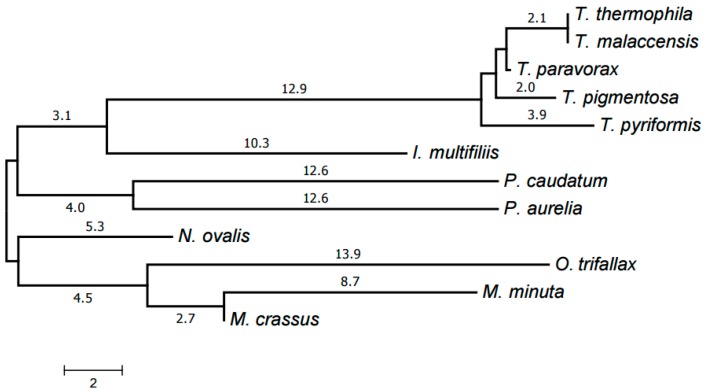
Evolutionary tree of mitochondrial chromosome structures. The tree was generated by the neighbor-joining method using distances between chromosome structures calculated as described in [19].

**Table 1 life-07-00009-t001:** Six highly conserved elements (HCEs) represented in the class Oligohymenophorea.

Species	1st Position	Sequence Fragments
**HCE 287**		
*T. malaccensis*	2984	AATTTAAATACTTGCATTAAGACTAATCGTGG
*T. pigmentosa*	2988	AATTTAAATACTTGCATTAAGACTAATCGTGG
*T. pyriformis*	2988	AATTTAAAAGCTTGCATTAATACTAATCTTGG
*T. thermophila*	2943	AATTTAAACACTTGCATTAAAACTAATCTTGG
**HCE 299**		
*T. malaccensis*	10523	GACACAC**CATA**TGAATTTAAATCATTAATAATTCAA
*T. pigmentosa*	10558	GATAAAC**CATA**TGAATTTAAATTATTACTAATTAAA
*T. pyriformis*	10589	GATAGAC**CATA**AGAATTTAAGTCATTATTTATTCAA
*T. thermophila*	10500	GATAGAC**CATA**TGAATTTAAATCATTATTAATTCAA
**HCE 290**		
*T. malaccensis*	4810	ATAAAATAAGTTCTAAAAATG**TGTA**TTAATTCCTTAAACATTTA
*T. paravorax*	5270	ATAAAATAAGTTCTTAATATA**TGTA**TAAATTCTTTAAACATTTA
*T. pigmentosa*	4811	ATAAAATATGTTCTAAAAATA**TGTA**TTAATTCTTTAAACATTTA
*T. pyriformis*	4839	ATAAAATAAGTTCTAAAAATA**TGTA**TCAATTCTTTAAACATTTA
**HCE 234**		
*T. malaccensis*	4788	TTTTTTTAAATATCTAAAAGTAATAAAATAAGTTCTAAA
*T. paravorax*	5248	TTTTTTTAAATATCTAAATGTTATAAAATAAGTTCTTAA
*T. pigmentosa*	4789	TTTTTTAAAATATCTAAAAGTTATAAAATATGTTCTAAA
*T. pyriformis*	4817	TTTTTTGATATATCTAAAAGTGATAAAATAAGTTCTAAA
*T. thermophila*	4756	TTTTTTTAAATATCTAAAAGTAATAAAATAAGTTCTAAA
**HCE 138**		
*I. multifiliis*	1364	TTTAGGTGCAGCTAT
*I. multifiliis*	47702	TATAGCTGCACCTAAAAAAAAAAAA
*T. malaccensis*	27009	AATAGCCGCACCTAAAAGAAAAAAATCTA
*T. paravorax*	26884	AATAGCTGCTCCAAAAAGAAAAAAATCAA
*T. pigmentosa*	26364	AATAGCCGCACCTAAAAGAAAAAAATCCA
*T. pyriformis*	26770	AATGGCCGCACCTAAAAGAAAAAAATCAA
*T. thermophila*	27061	AATAGCCGCACCTAAAAGAAAAAAATCTA
**HCE 315**		
*T. malaccensis*	26891	ATAA**CGTA**TTTACAATAAAAAAATAAT
*T. pigmentosa*	26211	TCAA**CGTA**TTTACAATAAAATAATAAA
*T. pyriformis*	26678	TTAA**CGAA**TTTACAATAAAAAAATAAA
*T. thermophila*	26921	TTAA**CGTA**TCTACAATAAAAAAATAAA

**Table 2 life-07-00009-t002:** Distribution of proteins in clusters and singletons. Three columns on the right specify the numbers of proteins encoded in the mitochondrion, nontrivial clusters, and singletons for each species.

Locus	Species	Proteins	Clusters	Singletons
NC_015981.1	*Ichthyophthirius multifiliis*	41	39	2
GQ903131.1	*Moneuplotes crassus*	29	25	4
GQ903130.1	*Moneuplotes minuta*	36	30	6
GU057832.1	*Nyctotherus ovalis*	35	13	22
JN383843.1	*Oxytricha trifallax*	99	31	68
NC_001324.1	*Paramecium aurelia*	46	41	5
NC_014262.1	*Paramecium caudatum*	42	41	1
NC_008337.1	*Tetrahymena malaccensis*	45	44	0
NC_008338.1	*Tetrahymena paravorax*	44	43	1
NC_008339.1	*Tetrahymena pigmentosa*	44	44	0
NC_000862.1	*Tetrahymena pyriformis*	44	44	0
NC_003029.1	*Tetrahymena thermophila*	45	44	0

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
