# Peer review of "Highly Conserved Elements and Chromosome Structure Evolution in Mitochondrial Genomes in Ciliates"

_life, 2017, doi:10.3390/life7010009_

Round 1

Reviewer 1 Report

This manuscript of Gershgorin et. al. indicates that even small non coding but highly conserved elements in mitochondrial genomes of ciliates can be used for phylogenitic analyses. It is interesting to see that this kind of analyses can result to a same phylogeny as with performed with conserved protein coding genes. However I have no doubts about there analyses there are major problems in the structure of the manuscript and the presentation of the results. Some of them are listed below.

The introduction is unstructured. The first part of the introduction (lines 29-41) is an exact copy of the abstract. I would like to advice the authors to rewrite the complete introduction, start with a general introduction and later focus on the research question.

Line 42: what is written here is true but the main job of mitochondria is producing ATP. (and line 42-45)

Line 43: “lack mitochondria” this is to easy. Many anaerobic protists contain mitochondrial remnants like hydrogenosomes or mitosomes. Until now it seems there is only one suspect of complete mitochondrial loss (Karnkowska et al., 2016, Current Biology 26, 1274–1284May 23, 2016) and here are still some questions that need to be answered.

Include some information about the background of the investigated ciliates. Euplotes was isolated from coastal waters, n. ovalis, anaerobic, lives in the guts of cockroaches…..

To my opinion also the conclusions need to be rewritten. Maybe a native English speaker can help?

Table 2: Why the superscript at Proteins? The remark below the table is almost identical to the information above this table.

There are a lot more little things but to my opinion the above mentioned organization of the manuscript should be done first.

Author Response

Dear reviewer,

we are grateful for your thorough and constructive criticism.

Our replies follow each comment.

1) The first part of the introduction (lines 29-41) is an exact copy of the abstract. 

The abstract was shortened by omitting a brief description of well-known phylogenetic methods. Only the main concepts of our study were preserved. Specifically, it relies on the algorithms recently developed by us; and a statement was proposed, which is so far confirmed for mitochondria of ciliates (in this report) as well as for chloroplasts of monocotyledons and rhodoplasts of red algae and protists (the latter will be covered in a separate paper). The statement is valid for certain organelles and taxa and the range of its applicability remains open. In the MS, some of identified HCEs have been analyzed (which is not an easy matter; and the analysis of all found HCEs is yet harder). Naturally, the content of the abstract is reproduced in the introduction.

2) The introduction is unstructured. I would like to advice the authors to rewrite the complete introduction, start with a general introduction and later focus on the research question. 

Corrected.

3) Line 42: what is written here is true but the main job of mitochondria is producing ATP (and line 42-45).

Corrected.

4) Line 43: “lack mitochondria” this is to easy. Many anaerobic protists contain mitochondrial remnants like hydrogenosomes or mitosomes. Until now it seems there is only one suspect of complete mitochondrial loss (Karnkowska et al., 2016, Current Biology 26, 1274–1284May 23, 2016) and here are still some questions that need to be answered. 

Corrected.

5) Include some information about the background of the investigated ciliates. Euplotes was isolated from coastal waters, n. ovalis, anaerobic, lives in the guts of cockroaches….. 

Corrected.

6) To my opinion also the conclusions need to be rewritten. 

Corrected. It is a brief reproduction of the obtained results; we find it difficult to significantly expand this section.

7) Maybe a native English speaker can help?

The whole text was checked by a local linguistic service.

8) Table 2: Why the superscript at Proteins? 

Corrected.

9) The table is almost identical to the information above this table. 

We assumed that the table caption should reflect its content irrespective of the text.

Reviewer 2 Report

This study continues previous works of Lyubetsky’s lab related to Highly Conserved Elements (HCE). In this study, I found an original idea that «at least for some organelles and taxa, the genome and HCE structures, despite themselves containing relatively little information, still adequately resolve the evolution of species». To make this hypothesis more solid one should make try to clarify for what organelles, for which taxa, etc. In this manuscript, the hypothesis is tested on mitochondrial genomes of Ciliophora. The authors constructed a few genomic trees and compare them with taxonomy and phylogeny of Ciliophora. The trees were constructed using HCE, some protein families, and chromosome rearrangements. These three genome characterizations are rather different, and algorithms of tree construction from these data are different as well. All these attitudes were greatly developed by this group, for example, in BMC Bioinformatics (2016). I hope that this group will bring us new, more extended results in the nearest future. Nevertheless, the results on Ciliophora are interesting and valuable by themselves. A special interest present a following finding: there are HCE in non-coding regions. Some of these HCE are putative promoter elements, and this finding is interesting as well. As usual in this field, produced trees are not identical to taxonomical classifications but the differences are valuable, as well (from my point of view). Development and verification of the abovementioned phylogenomic methods are interesting and promising.

There are a few minor comments.

1)      Formatting of some names of species is not correct. I propose to check all the names in the study and use correct format (italics) everywhere.

2)      In section 2.1 (lines 75-77) I would propose to mention a) using what method the tree was generated and b) why RAxML and the method were chosen. (The method is mentioned in the section Methods but here it should be explained the choice)

3)      Very similar to my previous comment I would relate to lines 169-170 (why MUSCLE) and 173-177 (why RAxML and what method, why PhyloBaes)

4)      I would propose to expand a bit section Conclusions (lines 209-215). “Good agreement” … with what?

In section 4 one can find SOME answers to the above questions. However, there are no explanations why ML in one case and NJ in another, etc.

Author Response

Dear reviewer,

we are grateful for your thorough and constructive criticism.

Our replies follow each comment.

1) Formatting of some names of species is not correct. I propose to check all the names in the study and use correct format (italics) everywhere.

Corrected.

2) In section 2.1 (lines 75-77) I would propose to mention a) using what method the tree was generated and b) why RAxML and the method were chosen. (The method is mentioned in the section Methods but here it should be explained the choice).

Corrected. RAxML is based on the maximum likelihood (ML) method; PhyloBayes, on the Bayesian inference (BI); and the neighbor-joining method (NJ), on the distance matrix. Unlike PhyloBayes and NJ, RAxML relies here on the matrix of characters composed of ones and zeros. Accordingly, we used NJ when the distance matrix was available; PhyloBayes was used when the supermatrix compiled from proteins was available as the most advanced method in this case, although a faster RAxML is also commonly used. 

3) Very similar to my previous comment I would relate to lines 169-170 (why MUSCLE) and 173-177 (why RAxML and what method, why PhyloBaes).

MUSCLE is widely used for multiple alignment; see also responses 2 and 5.

4) I would propose to expand a bit section Conclusions (lines 209-215). “Good agreement” … with what?

Corrected.

5) In section 4 one can find SOME answers to the above questions. However, there are no explanations why ML in one case and NJ in another, etc. 

Some reasoning was given above; to some extent the method selection was partly subjective.

Reviewer 3 Report

In this manuscript, the mitochondrial genomes in Ciliophora were analyzed and the evolutionary trees of the genomes were generated using highly conserved elements (HCEs), proteins clustered by an original method (protein families, PFs), and chromosome rearrangements (CRs). A comparison of these different approaches is of interest and should be continued. The authors’ approaches to the HCE, PF, and CR models and especially to the underlying algorithms are highly original; they were represented in two BMC Bioinformatics publications in 2016 (as well as in other cited publications). The manuscript can be considered as an extension of these studies. The identified HCEs without coding regions are discussed separately. Several potential promoters have been predicted, which is also of interest. A relatively small number of identified HCEs due to the short length of mitochondrial DNA does not allow certain conclusions about the applicability of the proposed method for generating phylogenetic trees and evolutionary scenarios based on the identification of HCEs, PFs, and CRs.

Quality of Presentation: High (minor technical corrections are required, i.e., many species names should be italicized in lines 59 to 71).

Scientific Soundness: Average (the trees built using different methods do not match due to the short size of mtDNA, which assumes the study should be extended to other organelles and taxa)

Interest to the readers: Average (the main assumption about the significance of using HCEs and CRs in phylogenetic studies requires the analysis of other organelles and taxa; it is advisable to explore the biological role of other HCEs identified as well as to search for other HCEs).

Previously developed complex and detailed algorithm is of great potential advantage.

Author Response

Dear reviewer,

we are grateful for your constructive criticism.

Species names were italicized as you suggested.

Round 2

Reviewer 1 Report

This manuscript has improved seriously. I still think the introduction could be improved however this part is a lot better than before. The abstract is now a real abstract and the introduction became a separate section. I would like to suggest to make a more fluent story of the introduction.